# Survival Prediction in Diabetic Foot Ulcers: A Machine Learning Approach

**DOI:** 10.3390/jcm12185816

**Published:** 2023-09-07

**Authors:** Alina Delia Popa, Radu Sebastian Gavril, Iolanda Valentina Popa, Laura Mihalache, Andreea Gherasim, George Niță, Mariana Graur, Lidia Iuliana Arhire, Otilia Niță

**Affiliations:** 1Faculty of Medicine, University of Medicine and Pharmacy “Grigore T Popa”, 700115 Iasi, Romania; alina.popa@umfiasi.ro (A.D.P.); laura.mihalache@umfiasi.ro (L.M.); andreea.gherasim@umfiasi.ro (A.G.); george.nita@umfiasi.ro (G.N.); lidia.graur@umfiasi.ro (L.I.A.); otilia.nita@umfiasi.ro (O.N.); 2Faculty of Medicine and Biological Sciences, University “Ștefan cel Mare” of Suceava, 720229 Suceava, Romania; graur.mariana@gmail.com

**Keywords:** diabetic foot ulcers, machine learning, University of Texas Staging System, Wagner–Meggitt classification, Saint Elian Wound Score System, mortality

## Abstract

Our paper proposes the first machine learning model to predict long-term mortality in patients with diabetic foot ulcers (DFUs). The study includes 635 patients with DFUs admitted from January 2007 to December 2017, with a follow-up period extending until December 2020. Two multilayer perceptron (MLP) classifiers were developed. The first MLP model was developed to predict whether the patient will die in the next 5 years after the current hospitalization. The second MLP classifier was built to estimate whether the patient will die in the following 10 years. The 5-year and 10-year mortality models were based on the following predictors: age; the University of Texas Staging System for Diabetic Foot Ulcers score; the Wagner–Meggitt classification; the Saint Elian Wound Score System; glomerular filtration rate; topographic aspects and the depth of the lesion; and the presence of foot ischemia, cardiovascular disease, diabetic nephropathy, and hypertension. The accuracy for the 5-year and 10-year models was 0.7717 and 0.7598, respectively (for the training set) and 0.7244 and 0.7087, respectively (for the test set). Our findings indicate that it is possible to predict with good accuracy the risk of death in patients with DFUs using non-invasive and low-cost predictors.

## 1. Introduction

In recent years, there has been a rapid increase in the prevalence of diabetes mellitus and its associated complications, making it the leading cause of morbidity and death. Diabetes mellitus (DM) is a significant problem in today’s world, as there has been a substantial increase in the prevalence of individuals affected by this condition, with the number of diagnosed cases more than tripling over the course of the last two decades [1]. According to current estimates, the global prevalence of diabetes in 2021 was around 537 million individuals. Projections indicate that this number is expected to rise to 643 million by 2030, and further increase to 783 million by 2045 [2]. Hence, diabetic foot and lower limb issues have a significant impact on a substantial number of individuals worldwide who have diabetes, resulting in a considerable burden of illness. In addition to the adverse effects on quality of life, persistent ulcers and amputations also impose significant financial burdens on healthcare systems. Therefore, the expenses associated with providing treatment for a diabetic patient with a foot ulcer are 5.4 times more during the first year and 2.6 times greater in the subsequent year [1]. When considering the debilitating nature of this complication, it is important to acknowledge the accompanying social disengagement and inability to work. These factors contribute to further losses, both at the individual and collective level, in terms of public health [2].

The diabetic foot is characterized by three significant pathophysiological pillars: neuropathy, ischemia, and infection. These factors, albeit intricate in nature, play a crucial role in the development and progression of diabetic foot complications. The incidence of diabetic neuropathy in adults with diabetes ranges from 16% to 87% [3,4]. Additionally, individuals with diabetes mellitus (DM) are more likely to have peripheral arterial disease owing to the unique characteristics of atherosclerosis in DM [5].

The primary focus should be on preventing the onset of diabetic foot complications. This objective may be accomplished by implementing effective disease control measures and promptly identifying individuals at high risk, such as those with peripheral neuropathy, peripheral vascular disease, foot abnormalities, and the presence of calluses. The identification of specific risk factors among the predetermined predisposing factors is of utmost importance due to the significant rise in the occurrence of diabetic foot infections. The identification of the most significant risk factors among various variables is crucial for determining diagnostic and therapeutic protocols. This consideration plays a vital role in health management, treatment priority, and the enhancement of the patient’s quality of life.

The probability of developing a diabetic foot ulcer (DFU) in individuals with DM fluctuates between 15% and 25% throughout the course of their lifetime, with a majority of these individuals needing amputation within four years of the initial diagnosis [6]. In comparison to those without diabetes, patients with diabetes mellitus exhibit a significantly elevated incidence of major amputations, with rates ranging from 30 to 40 times higher. Therefore, it may be inferred that diabetic foot syndrome accounts for around 80% of non-traumatic amputations [7,8,9]. According to estimates, there is a global occurrence of limb amputations every 30 s, mostly attributed to lower limb peripheral artery disease [4]. In addition to exerting an effect on the overall quality of life, diabetic foot ulceration also has a significant impact on death rates. Therefore, it can be shown that the mortality rate among individuals diagnosed with diabetes mellitus and suffering from ulcers on the lower extremities is more than twice as high when compared to patients with diabetes mellitus but without ulcers. This conclusion is consistent regardless of variables such as age, diabetes type, or duration of the disease [10]. Patients diagnosed with diabetes mellitus (DM) and suffering from foot ulcers exhibit a significantly reduced survival rate of 5 years, amounting to 43%. This survival rate is notably lower when compared to those without diabetes who also have ulcers, with a survival rate of 56%. Furthermore, when compared to the whole population, the survival rate of patients with DM and foot ulcers is even more pronouncedly lower, at 68% [11]. Furthermore, it has been estimated that the five-year death rate after amputation ranges from 39% to 68%, which is equivalent to the life expectancy of individuals with aggressive cancer or severe stages of congestive heart failure [7].

Clinical systems that include risk factors for DFU recurrence have been developed to classify diabetic foot conditions, such as the University of Texas Staging System for Diabetic Foot Ulcers; the Perfusion, Extent, Depth, Infection, and Sensation (PEDIS) Classification System and Score; the International Working Group on the Diabetic Foot (IWGDF) Guidelines on the Prevention and Management of Diabetes-Related Foot Disease; and the SINBAD (Site, Ischemia, Bacterial Infection, Area, and Depth) system [12,13]. However, there remains a lack of adequate studies that have explored the relationship between the severity of DFUs and outcomes such as ulcer-free survival, hospitalization, and mortality. This indicates a need for further research in this area [14]. Hence, it is necessary to conduct more research in order to address this gap in information and improve our capacity to reliably forecast death rates in patients with diabetic foot ulcers [13,14,15].

Our study proposes the first artificial intelligence (AI)/machine learning (ML) model to predict long-term mortality (5 and 10 years) in diabetic patients with foot ulcers. Enhancing mortality prediction in diabetes patients with foot ulcers will finally lead to the intensified control of modifiable risk factors and improved long-term prognostic. 

## 2. Materials and Methods

### 2.1. Study Design

A single-center observational retrospective study was conducted, including a total of 659 individuals diagnosed with both type 1 and type 2 diabetes mellitus with associated DFUs. The Diabetes, Nutrition, and Metabolic Diseases Clinic, within the Emergency Clinical Hospital “Sf. Spiridon” in Iași, functions as a regional center for patients from eight counties in the north-eastern area of Romania. All individuals included in this study were admitted to the clinic over the period spanning from 2007 to 2017. The patients were monitored until the year 2022, until their death, or until the last date documented in the computerized case system. The patients who were hospitalized at our regional facility were either referred via the emergency department, by their diabetologist or general practitioner, or were transferred from other clinics. Inclusion criteria were the presence of diabetes mellitus and the presence of a discharge diagnostic code relevant for DFU according to the International Statistical Classification of Diseases, 10th Revision (ICD-10) [16]. Cases were eliminated from the research if concurrent disorders that may potentially impact the clinical and biological data were present. These conditions encompassed a history of prior foot ulcers or amputations, ongoing neoplastic diseases, end-stage respiratory conditions, hepatic failure, end-stage renal disease, heart failure at stages III–IV according to the New York Heart Association classification, cognitive impairment, psychiatric disorders, and collagen vascular disorders. 

A comprehensive database was created in collaboration with the hospital’s statistics service, encompassing patients whose discharge records contained ICD-10 codes associated with diabetic foot pathology. A thorough investigation was conducted to determine the discharge diagnosis. Patients who did not have a diagnosis linked to diabetic foot ulcers (DFUs) were excluded from the database.

### 2.2. Data Acquisition

The data gathered for the study included numerous continuous variables: age, disease duration (years), date of death, year of admission, duration of hospitalization (days), readmissions, The University of Texas Staging System for Diabetic Foot Ulcers, the Saint Elian Wound Score System (SEWSS), hemoglobin (g/dL), hematocrit, white blood cells/mm^3^, neutrophils/mm^3^, plateles/mm^3^, fibrinogen (mg/dL), C reactive protein (CRP) mg/dL, glucose levels (mg/dL), HbA1c mmol/mol, urea (mg/dL), creatinine (mg/dL), alkaline reserve (mmol/L), sodium (mmol/L), potassium (mmol/L), aspartate aminotransferase (ASAT) (U/L), alanine aminotransferase (U/L), total proteins (g/L), albumin (g/L), sideremia (μg/dL), ferritin (ng/mL), total cholesterol (mg/dL), high-density lipoprotein—HDL (mg/dL), low-density lipoprotein—LDL (mg/dL), triglycerides (mg/dL), uric acid (mg/dL), and glomerular filtration rate (mL/min/1.73 m^2^).

Numerous categorical variables were also collected: sex, the presence of diabetic retinopathy, diabetic nephropathy, chronic kidney disease, cardiovascular disease, peripheral artery disease, sensorimotor polyneuropathy, autonomic neuropathy, Charcot foot, hypertension, dyslipidemia, hyperuricemia, obesity, metabolic decompensation at admission (ketosis, ketoacidosis), treatment with metformin/insulin/or other classes of oral or injectable antidiabetics prior to admission, Wagner–Meggitt classification, variables from the Saint Elian Wound Score System (the primary location of the lesion, topographic aspects of the lesion, the number of affected areas, ischemia, classification of infection, edema, neuropathy, the area of the lesion, depth of lesion, wound healing phase), cellulite, fever, radiography osteolysis, and wound secretion.

### 2.3. Outcome Definition

The outcomes to be predicted by our models are binary variables that estimate whether the patient will die in the 5 or 10 years following hospitalization.

### 2.4. Preprocessing

Documented continuous variables were normalized in the range [0,1]. 

Records with missing values were removed, resulting in a database comprising 635 total records. 

### 2.5. Feature Selection

Since the continuous parameters did not follow a normal distribution, we employed the Kruskal–Wallis rank sum test to assess potential associations between these continuous variables and the outcome. In cases where two of the selected continuous variables displayed high intercorrelation with a Pearson coefficient of ≥0.9, one of them was excluded.

Similarly, we employed Pearson’s Chi-squared test of independence to investigate whether any connections existed between each categorical parameter and the outcome.

Subsequently, the continuous and categorical parameters that demonstrated a statistically significant relationship with the outcome (*p* < 0.05), as determined by the Kruskal–Wallis and Pearson’s Chi-squared tests, were chosen as predictors for the machine learning models.

### 2.6. Development of the ML Models

The initial database of 635 UC patient records was randomly divided into two subsets: a training set consisting of 508 records (80%) and a test set comprising 127 records (20%). 

Two multilayer perceptron (MLP) classifiers were constructed. The models were developed using the caret::train function within R Studio (version 1.4.1106). To safeguard against overfitting, we implemented a ten-fold cross-validation approach. To address the challenge posed by imbalanced outcome classes, we integrated the synthetic minority over-sampling technique (SMOTE) into the caret::train function.

The first MLP model was developed to predict whether the patient will die in the next 5 years after the current hospitalization based on the selected variables. The second MLP classifier was built to estimate whether the patient will die in the following 10 years based on all the selected predictors. 

We assessed the performance of the ML models by measuring their classification accuracy on both the test and training datasets. Additionally, we calculated other performance metrics including the area under the receiver operating characteristic curve (AUC), sensitivity, specificity, positive predictive value (PPV), and negative predictive value (NPV).

### 2.7. Variable Importance

For both developed models, we applied the varImp function (caret library) in R to determine the importance of each variable in predicting the outcome. For MLP models, the varImp function will take as an argument the resulting MLP model and will sort the importance of the variables based on a “filter” approach (ROC curve analysis).

### 2.8. Ethical Considerations

The present research received authorization from the Ethics Commission of the University of Medicine and Pharmacy “Gr. T. Popa”, Iaşi, under the reference number 2324/19.01.2017. The retrospective, observational nature of the investigation precluded the solicitation of informed consent from patients. The use of participants’ identity data for any other purposes has not been and will not be conducted.

## 3. Results

After excluding records with missing values, the study’s database comprised 635 diabetic patients, with 435 (69%) being male and 224 (31%) being female. The ages of the participants ranged from 19 to 89 years. Table 1 provides an overview of the selected laboratory findings for all patients, as well as for each outcome class (those deceased in 5 years and those deceased in 10 years). Continuous variables are presented as median values (interquartile range), while categorical variables are depicted by the count of occurrences in each category. It is worth noting that the study’s outcome classes exhibit an imbalance in size, as evident from Table 1. This disparity prompted the utilization of SMOTE in the development of the models. 

To identify the continuous and categorical variables significantly associated with the study’s outcomes, we applied Kruskal–Wallis rank sum and Chi-square tests, with the corresponding *p*-values displayed in Table 1.

The feature selection process consequently pinpointed the following predictors for utilization in training the 5-year prediction model (*p* < 0.05): age, hemoglobin, creatinine, urea, duration of hospitalization, Texas classification, ischemia from the SWESS, neuropathy from the SWESS, glomerular filtration rate, cardiovascular disease, SEWSS score, Wagner classification, topographic aspects of the lesion from the SWESS, hypertension, diabetic nephropathy, peripheral artery disease, hyperuricemia, and depth of lesion from the SWESS. For the 10-year prediction model, the same set of predictors was selected, with the exception of peripheral artery disease and hyperuricemia.

There were no high inter-correlations (Pearson coefficient ≥ 0.9) between the selected continuous variables (Figure 1).

Using the variables identified as predictors through the feature selection process, we proceeded to train two ML models.

We developed the 5-year mortality model to predict whether the patient will die in the following 5 years after hospitalization. The final values used for the model were: three hidden layers with four neurons in the first layer, one neuron in the second layer, and five neurons in the third layer. Table 2 presents the performance metrics overview of the classifier.

The ROC curve obtained by the 5-year model both on the training and test sets is illustrated in Figure 2. 

The variable importance determined by the varImp function is graphically represented in Figure 3.

The 10-year mortality classifier was built to estimate whether the patient will die within 10 years after hospital admission. Table 3 displays the performance metrics achieved by this classifier. The 10-year prediction model contains three hidden layers with five neurons in the first layer, five neurons in the second layer, and four neurons in the third layer. Table 3 presents the performance metrics overview of the classifier.

Figure 4 shows the ROC curve that illustrates the performance of the 10-year mortality classifier on both the training and test sets. 

The variable importance corresponding to the 10-year prediction model is graphically illustrated in Figure 5.

## 4. Discussion

Our research has proposed an ML method that can predict long-term mortality in diabetic patients with foot ulcers. To our knowledge, this is the first study in the literature to propose an ML model aimed to estimate long-term mortality for diabetic patients with diabetic foot ulcers. Our findings indicate that it is possible to predict with good accuracy whether the patient will die in the following 5 or 10 years using non-invasive and low-cost clinical and biological predictors.

Initially, the study undertook the process of feature selection, which led to the identification of several non-invasive predictors related to the 5- and 10-year mortality outcome: age, hemoglobine, creatinine, urea, length of hospitalization, The University of Texas Staging System for Diabetic Foot Ulcers score, ischemia from the Saint Elian Wound Score System (SEWSS), neuropathy from the SEWSS, glomerular filtration rate, cardiovascular disease, SEWSS score, the Wagner–Meggitt classification of foot ulcers grade, topographic aspects of the lesion from the SEWSS, hypertension, diabetic nephropathy, peripheral artery disease, hyperuricemia, and depth of lesion from the SEWSS. This is in accordance with the scientific literature as, to date, many of these variables were studied as mortality predictors in diabetes mellitus patients [17,18,19,20]. However, our paper is the first to consider combining the predictive power of all these variables by developing an AI/ML classifier. No significant inter-correlations (Pearson coefficient > 0.9) were observed among the chosen continuous variables. The accuracy of the 5-year model was 0.7717 for the training set and 0.7244 for the test set. Similarly, the accuracy of the 10-year model was 0.7598 for the training set and 0.7087 for the test set. The area under the curve, sensitivity, specificity, negative predictive value, and positive predictive value were all deemed satisfactory for both models.

The predictive model for 5-year mortality, which aimed to determine the likelihood of patient death within five years after hospitalization, incorporated the following variables in descending order of significance: age, The University of Texas Staging System for Diabetic Foot Ulcers score, the Wagner–Meggitt classification of foot ulcers grade, the Saint Elian Wound Score System (SEWSS), glomerular filtration rate, the presence of cardiovascular disease, creatinine serum value, urea serum value, the depth of lesion from the SEWSS, duration of hospitalization, the presence of peripheral artery disease, the value of hemoglobin, topographic aspects of the lesion, diabetic nephropathy, the presence of high blood pressure, the presence of ischemia according to the SEWSS, and the presence of hyperuricemia. The predictors included in the 10-year mortality model, listed in order of importance, were age, the presence of neuropathy, glomerular filtration rate, The University of Texas Staging System for Diabetic Foot Ulcers score, serum urea level, the Wagner–Meggitt classification of foot ulcers grade, creatinine serum level, presence of diabetic nephropathy, presence of cardiovascular disease, duration of hospitalization, the Saint Elian Wound Score System (SEWSS), presence of high blood pressure, depth of lesion, hemoglobin value, topographic aspects of the lesion, and the presence of ischemia according to the SEWSS. The disparity between the 5-year and 10-year prediction models lies in the inclusion of peripheral artery disease and hyperuricemia in the former, while the latter includes the grading of diabetic neuropathy based on the SEWSS. 

The findings of our study provide empirical support for the need to adopt a multidisciplinary strategy in the management of patients with diabetic foot ulcers (DFUs). The implementation of a multidisciplinary treatment paradigm for individuals with diabetic foot conditions has the potential to provide positive results and enhance overall health [21]. A study conducted on 1428 patients concluded that DFU represents a major cause of major comorbidities and mortality. Within a span of five years after the occurrence of DFUs, individuals diagnosed with DFUs had a significantly greater absolute risk of death, with rates of 107.7 per 1000 person-years compared to 33.7 per 1000 person-years [22].

Diabetic chronic kidney disease (CKD) leads to gradual injuries of the skeletal and vascular systems, giving rise to a complex condition referred to as chronic kidney disease-mineral bone disorder (CKD-MBD) [23,24]. The development of these features may begin as early as stage 2 chronic kidney disease (CKD) and often becomes apparent once the estimated glomerular filtration rate (eGFR) drops below 60 mL/min/1.73 m^2^, which corresponds to stage 3 CKD [23]. In a study on patients with diabetic neuropathic feet, specific biomarkers, such as the presence of peripheral neuropathy, the calcification of pedal vessels, and a decrease in the metatarsals’ capacity to withstand bending forces, supported the presence of CKD-MBD syndrome. Furthermore, the frequency of single-foot biomarkers, as well as combinations of two- and three-foot biomarkers, exhibits an upward trend as chronic kidney disease (CKD) advances through its stages [23].

Ischemic heart disease is often seen in individuals with DFU, with a prevalence rate of up to 60% of cases [25]. A recent meta-analysis reported a pooled prevalence rate of 26% for this comorbidity [26]. Moreover, individuals with diabetic foot ulcers (DFUs) had a significantly increased likelihood of experiencing a fatal myocardial infarction (RR = 2.22, 95% CI 1.09–4.53). However, cardiovascular disease (CVD) mortality exhibited a comparable percentage across individuals with diabetic foot ulcers (DFUs) and those without DFUs [27]. In another study, the reported 5-year death rate in people with diabetic foot ulcers (DFUs) and ischemic heart disease (IHD) was found to be 40% [28]. A study including 655 individuals with diabetic foot ulcers (DFUs) showed that using a proactive strategy to manage cardiovascular risks resulted in a significant decrease in 5-year death rate, from 48.0% to 26.8% [29]. 

The findings of our study underscore the need to adopt an interdisciplinary strategy when managing patients with diabetic foot ulcers (DFUs). Our analysis revealed a significant association between fatality risk and both the severity of ulcers and the coexistence of cardiovascular and renal chronic conditions. The enhancement of survival rates for patients demands collaborative efforts among diabetologists, cardiologists, nephrologists, and surgeons.

In individuals diagnosed with diabetes and a foot ulcer, it is recommended by guidelines to utilize either the SINBAD (Site, Ischaemia, Bacterial infection, Area and Depth) system or the WIfI (Wound, Ischaemia, foot Infection) system for effective communication among healthcare providers. It is important to note that in both cases, the specific components of these systems should be described individually rather than relying solely on a cumulative score [13]. The ‘Guidelines on the categorization of foot ulcers in individuals with diabetes (IWGDF 2023 update)’ states that there is no current approach that can be suggested for predicting the outcome of an ulcer in a particular individual [13]. In our study, risk factors associated with the 5- and 10-year mortality rate were characteristics of foot ulcers, such as the depth of lesion from the SEWSS, the presence of ischemia or neuropathy according to the SEWSS and topographic aspects of the lesion, but also the scores according to the University of Texas Staging System for Diabetic Foot Ulcers score, the Wagner–Meggitt classification of foot ulcers, and the Saint Elian Wound Score System (SEWSS). Consistent findings were seen across many studies that used survival analysis. Schofield et al. [30] examined the relationship between wound topography and the risk of mortality. The findings of this study revealed that hind foot ulcers were identified as an independent risk factor for death. In retrospective research [31], the reported overall survival rate at five years following the occurrence of DFU infection was 49.7% (95% CI 44.8–54.6%). The survival rate of patients without significant amputation was shown to be negatively impacted by the presence of ischemia, older age, and a higher level of C-reactive protein [31]. The existing literature on the associations between survival outcomes and various staging systems for diabetic foot ulcers (DFUs) has shown inconsistent findings. The University of Texas Staging System for Diabetic Foot Ulcers score was not a significant predictor of mortality in some research [32,33], while in a study conducted by Amadou et al. [34], 5-year-mortality was associated with age, diabetes duration, dialysis, PEDIS perfusion grade, PEDIS sensation grade, University of Texas wound classification stages C and D, and ulcer duration during the first year of follow-up. Contradictory findings were seen in investigations examining the association between mortality and the Wagner–Meggit classification [19,32,35] and SWESS [14]. Rubio et al. [19] performed a recent retrospective analysis whereby they found an independent association between mortality and the SINBAD rating system. There was a negative correlation between the progression of the WIfI stage and the amputation-free survival rate. Nevertheless, the study found no significant disparity in the overall survival rate across different WIfI grades [36]. The findings from our analyses, as well as the existing literature, highlight the significance of conducting a thorough evaluation of DFUs in order to assess the long-term outcomes for patients. Additionally, these results underscore the necessity for additional research aimed at developing improved classification systems or algorithms that can effectively stratify the long-term prognosis of individuals with DFU.

In a previous study based on conventional survival analysis, we showed that patients with deep ulcers involving the tendon, joint, or bone had a higher mortality risk [20]. In this paper, we passed beyond traditional modeling. Due to the ability of neural network models (such as MLPs) to take into account complex interactions between variables and construct a non-linear prediction model, neural networks provide a more versatile approach to predicting survival time compared to conventional techniques. Moreover, randomness is the standard for ML models to control for confounding variables. At present, ML methods are based on random initialization and various types of random choices during learning that control the impact of confounders. Unlike our previous paper, which did not identify the SEWSS as a relevant predictor of mortality [20], in the present study, the SEWSS proved to be an important predictor for the 5-year mortality estimation model. However, it is a less relevant predictor for the 10-year mortality classifier. Therefore, our paper emphasizes that stratifying survival analysis for discrete time intervals may produce surprising information and conclusions that are otherwise inaccessible from a conventional, continuous survival curve analysis.

### Strength and Limitations

A retrospective study was performed on a consecutive sample of patients hospitalized at the Diabetes, Nutrition, and Metabolic Diseases Clinic at “Sf. Spiridon” Hospital, Iaşi. The chosen study strategy facilitated the collection of data from a significant sample size of patients over a lengthy period of time. This is the first examination using machine learning techniques to evaluate the long-term mortality rates among individuals with diabetic foot ulcers (DFUs). The findings of our research provide further empirical evidence to reinforce the prediction of extended mortality in persons diagnosed with diabetic foot ulcers (DFUs), therefore enhancing the current corpus of academic literature on this particular subject.

The research was conducted at a single facility, namely “Sf. Spiridon” Hospital in Iaşi, which serves a wide geographical area in the north-eastern part of Romania. This hospital has departments specializing in general and vascular surgery, which allowed for the mitigation of selection bias. One primary constraint of our dataset is that the independent test set originates from the same center. Consequently, it is essential to conduct comprehensive external validation using data from many centers in future studies. One further concern is the unequal distribution of result classes, which has the potential to introduce computation biases. Nevertheless, efforts have been made to mitigate these biases via the use of the synthetic minority over-sampling technique (SMOTE) function in the R programming language.

One of the primary limitations of our study was the failure to include the specific therapeutic approach, whether medical or surgical, used for the treatment of diabetic foot ulcers (DFUs).

Furthermore, the use of a prospective study design would enhance the appeal of our research by allowing for the incorporation of additional variables that were previously unavailable to us. Furthermore, the implementation of surveys at multiple intervals would provide a more accurate assessment of the influence of glycemic control on our sample of patients.

In addition, the analysis of the association between DFUs and the cause of death was not conducted due to the limited availability of electronic case histories, which did not include this information. The Public Health Department in our area regularly communicates the registration of deceased patients, mitigating the potential for reporting bias in mortality data.

## 5. Conclusions

Our findings indicate that it is possible to predict with good accuracy the risk of death in patients with DFU in the following 5 or 10 years using non-invasive and low-cost clinical and biological predictors. Our study emphasizes the significance of a comprehensive strategy in managing patients with diabetic foot ulcers (DFUs), including thorough local examination, the consideration of comorbidities, and the identification of risk factors. Such an approach is best achieved within the framework of a multidisciplinary team.

## Figures and Tables

**Figure 1 jcm-12-05816-f001:**
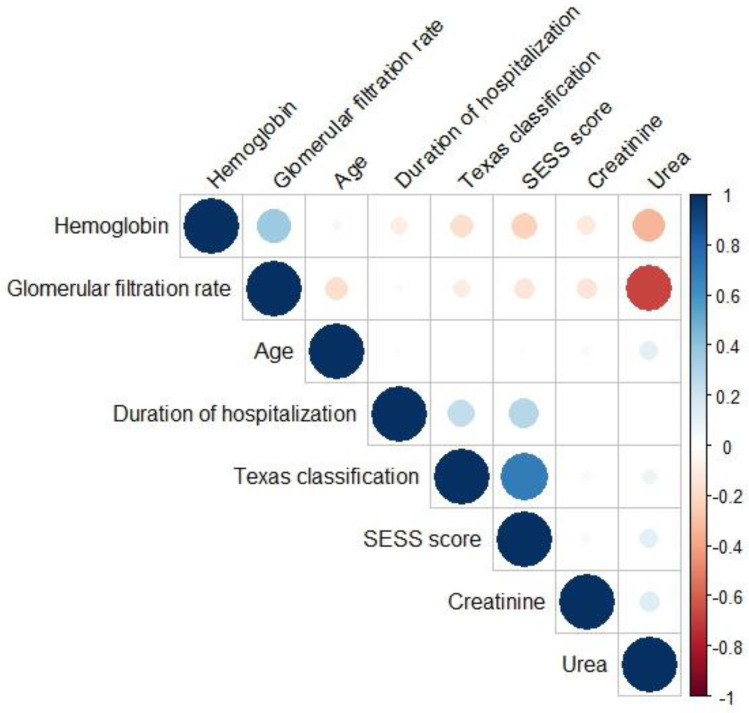
Pearson correlation heatmap between the selected continuous variables.

**Figure 2 jcm-12-05816-f002:**
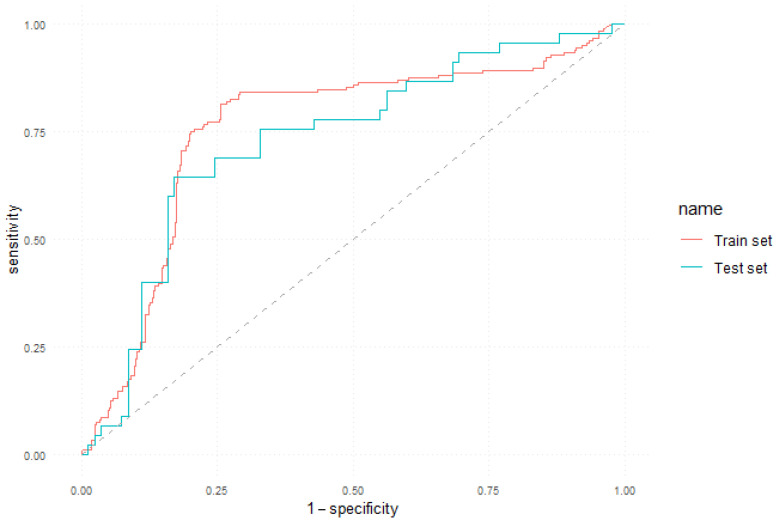
ROC curve of the 5-year prediction model.

**Figure 3 jcm-12-05816-f003:**
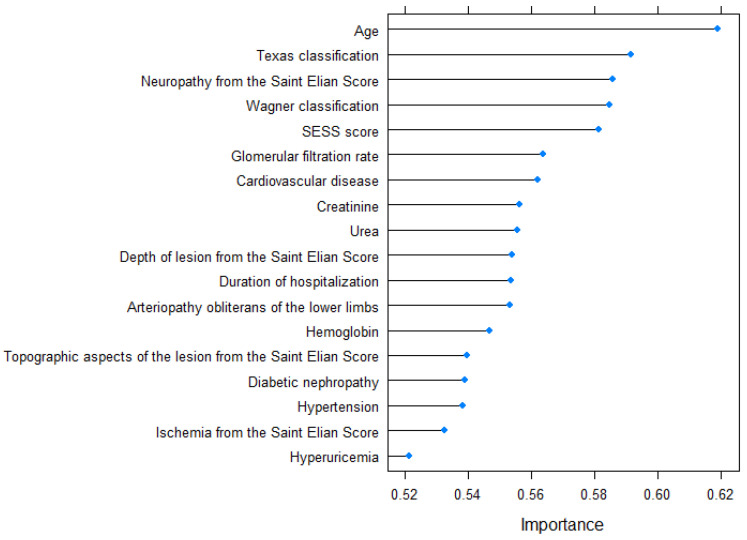
Variable importance for the 5-year prediction model.

**Figure 4 jcm-12-05816-f004:**
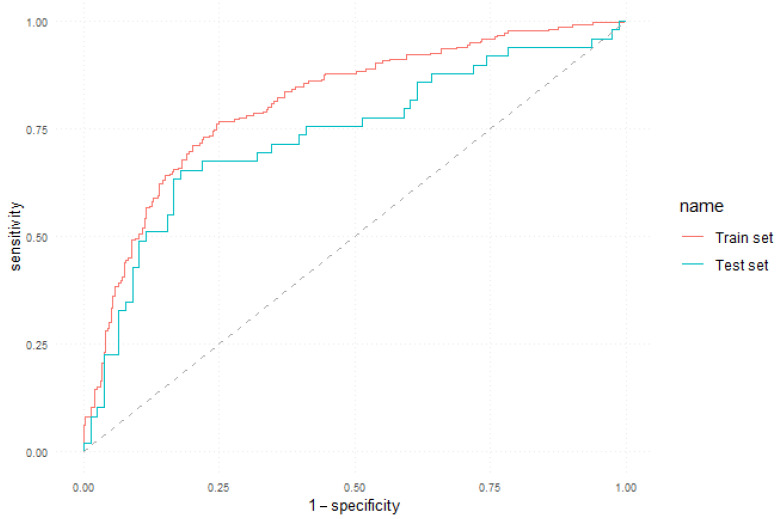
ROC curve of the 10-year prediction model.

**Figure 5 jcm-12-05816-f005:**
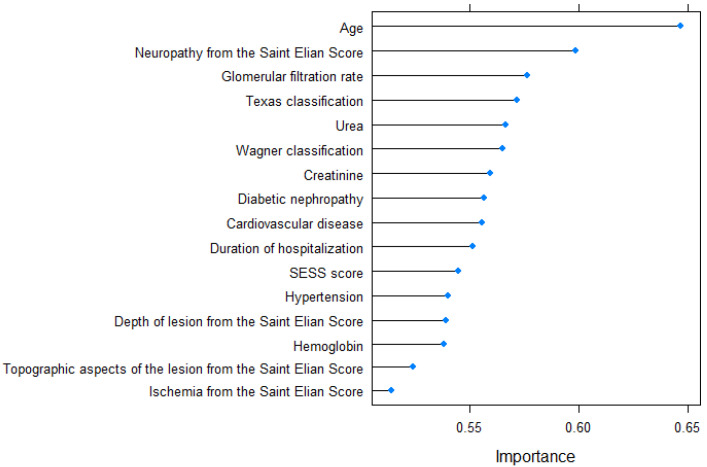
Variable importance for the 10-year prediction model.

**Table 1 jcm-12-05816-t001:** The selected parameters and gender for all patients and each outcome category.

	All	5 Years		10 Years	
		Deceased	Correlation with the 5-Year Outcome (*p*)	Deceased	Correlation with the 10-Year Outcome (*p*)
Number of records	635	221	-	263	-
Gender (male:female)	418:217	140:81	0.272	164:99	0.083
Age (years)	61 (14)	65 (15)	<0.001	65 (15)	<0.001
Hemoglobin (g/dL)	12.6 (2.6)	12.4 (2.9)	0.02	12.4 (2.95)	0.049
Creatinine (mg/dL)	0.99 (0.48)	1.04 (0.55)	0.011	1.03 (0.545)	0.006
Urea (mg/dL)	44 (27)	48 (31)	0.001	48 (30.5)	<0.001
Duration of hospitalization (days)	16 (13)	17 (15)	0.013	17 (14.5)	0.015
Texas classification	7 (6)	8 (9)	<0.001	8 (8.5)	<0.001
Ischemia from the SEWSS (absent:mild:moderate:severe)	230:196:87:122	80:51:32:58	<0.001	110:52:35:66	<0.001
Neuropathy from the SEWSS (Absent: Protective sensation diminished: Loss of protective sensation: Charcot neuro-osteoartropathy)	25:238:369:3	7:64:148:2	0.002	9:74:178:2	<0.001
Glomerular filtration rate (mL/min/1.73 m^2^)	73 (37.5)	69 (43)	0.002	68 (43.5)	0.002
Cardiovascular disease (yes:no)	256:379	114:107	<0.001	130:133	<0.001
SEWSS score	16 (5)	16 (5)	<0.001	16 (5)	0.022
Wagner–Meggitt classification (0:1:2:3:4:5)	9:208:186:95:114:23	0:61:61:39:52:8	0.007	3:70:81:44:57:8	0.029
Topographic aspects of the lesion from the SEWSS (Dorsal or plantar: Lateral or medial: Two or more)	516:50:69	172:15:34	0.013	209:16:38	0.013
Hypertension (yes:no)	466:169	174:47	0.015	206:57	0.013
Diabetic nephropathy (yes:no)	205:430	84:137	0.021	104:159	0.001
Peripheral artery disease (no:stage I:stage II A or IIB:stage III:Stage IV Leriche-Fontaine)	402:8:41:9:175	125:2:18:5:71	0.028	156:3:21:5:78	0.239
Hyperuricemia (yes:no)	53:582	22:199	0.036	24:239	0.25
Depth of lesion from the SEWSS (Superficial:Deep ulcer:All layers)	225:209:201	65:76:80	0.039	80:94:89	0.0522

**Table 2 jcm-12-05816-t002:** The performance metrics of the 5-year prediction model.

	Training Set		Test Set	
	Actual Values		Actual Values	
Predicted values	Survived	Deceased	Survived	Deceased
Survived	258	42	61	14
Deceased	74	134	21	31
Accuracy	0.7717		0.7244	
95% CI	(0.7326, 0.8075)		(0.6381, 0.7999)	
*p* value	<0.001		0.037	
Sensitivity	0.7771		0.7439	
95% CI	(0.7293, 0.8186)		(0.64, 0.826)	
Specificity	0.7614		0.6889	
95% CI	(0.6932, 0.8183)		(0.5433, 0.8047)	
PPV	0.86		0.8133	
95% CI	(0.8162, 0.8947)		(0.7107, 0.8854)	
NPV	0.6442		0.5962	
95% CI	(0.5771, 0.7061)		(0.4607, 0.7184)	
AUC	0.755		0.73	

AUC = area under the receiver operating characteristic curve; PPV = positive predictive value; NPV = negative predictive value.

**Table 3 jcm-12-05816-t003:** The performance metrics of the 10-year prediction model.

	Training Set		Test Set	
	Actual Values		Actual Values	
Predicted values	Survived	Deceased	Survived	Deceased
Survived	234	62	57	16
Deceased	60	152	21	33
Accuracy	0.7598		0.7087	
95% CI	(0.7203, 0.7964)		(0.6215, 0.7859)	
*p* value	<0.001		0.01675	
Sensitivity	0.7959		0.7308	
95% CI	(0.7462, 0.838)		(0.6232, 0.8166)	
Specificity	0.7103		0.6735	
95% CI	(0.6462, 0.7669)		(0.5338, 0.7879)	
PPV	0.7905		0.7808	
95% CI	(0.7406, 0.833)		(0.6732, 0.8603)	
NPV	0.7170		0.6111	
95% CI	(0.6529, 0.7733)		(0.4779, 0.7296)	
AUC	0.8095		0.7339	

AUC = area under the receiver operating characteristic curve; PPV = positive predictive value; NPV = negative predictive value.

## Data Availability

Data used in this study will be available from the corresponding authors upon request.

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
