# Peer review of "Survival Prediction in Diabetic Foot Ulcers: A Machine Learning Approach"

_jcm, 2023, doi:10.3390/jcm12185816_

Round 1

Reviewer 1 Report

Nice study, well written.

The main deficiency of this article is that no data given for this  635   patient

about how the foot ulcer treated. Medical therapy or surgical, since that can another parameter affecting survival.

Author Response

Reviewers’ Comments to Authors:

Reviewer: 1

Nice study, well written.

Answer 0.

We would like to thank the Reviewer and the Editorial board for all the positive remarks regarding our work. We are delighted to hear that the Reviewer observed the quality of the manuscript and the in-depth analysis of the subject.

Q1. The main deficiency of this article is that no data given for this  635 patient about how the foot ulcer treated. Medical therapy or surgical, since that can another parameter affecting survival.

Answer 1:

We appreciate your insightful perspective. A multidisciplinary team made up of diabetologists, general surgeons, and vascular surgeons manages patients at our hospital. The ideal approach for treating neuropathic DFUs is conservative therapy, which includes treating the infection, surgically removing necrotic tissue and the callus that surrounds it, draining abscesses, and offloading. The multidisciplinary team decides conservative treatment or amputation when osteomyelitis is present in order to save as much of the limb as feasible. AngioCT of the lower leg and aorta is conducted in the presence of PAD to determine the kind of revascularization and the degree of amputation, if necessary. As you pointed out, one of the main shortcomings of our research is that these characteristics were not taken into account when determining the patients' chances of survival. This aspect was added in our manuscript in the Strength and limitations section:

“One of the primary limitations of our study was the failure to include the specific therapeutic approach, whether medical or surgical, used for the treatment of diabetic foot ulcers (DFUs).”

Reviewer 2 Report

Alina et al, developed a machine learning model to predict long-term mortality in  patients with diabetic foot ulcers (DFUs). They reported two MLP classifiers for prediction of  mortality associated with  DFUs for 5 and 10 years. The study is interesting however few limitation are there .

Major: information related to death is required to validate mortality related to DFUs

Minor: another cohort and better from other hospital could improve the finding

Minor editing of English language required.

Author Response

Reviewer: 2

Alina et al, developed a machine learning model to predict long-term mortality in  patients with diabetic foot ulcers (DFUs). They reported two MLP classifiers for prediction of  mortality associated with  DFUs for 5 and 10 years. The study is interesting however few limitation are there .

Answer 0.

We would like to thank the Reviewer for the remarks regarding our work. We assure the Reviewer that we have read carefully the suggestions from this decision letter and tried our best to improve the quality of the document accordingly.

Q1. Major: information related to death is required to validate mortality related to DFUs.

Answer 1:

Thank you for pointing this out. We mentioned this aspect in the Strength and limitations paragraph:

“In addition, the analysis of the association between DFUs and the cause of death was not conducted due to the limited availability of electronic case histories, which did not include this information. The Public Health Department in our area regularly communicates the registration of deceased patients, so mitigating the potential for reporting bias in mortality data.” (lines 410-411)

Q2. Minor: another cohort and better from other hospital could improve the finding.

Answer 2:

Thank you for your valuable comment. We mentioned this aspect in the Strength and limitations section:

“The research was done at a single facility, namely "Sf. Spiridon" Hospital in IaÅŸi, which serves a wide geographical area in the north-eastern part of Romania. This hospital has departments specializing in general and vascular surgery, which allowed for the mitigation of selection bias. One primary constraint of our dataset is that the independent test set originates from the same center. Consequently, it is essential to do comprehensive ex-ternal validation using data from many centers in future studies.” (lines 392-397)

Reviewer 3 Report

I found the paper interesting and easy to read. Some small changes should be made.

1.       The author say in line  155  “As the continuous parameters did not exhibit a normal distribution, we performed  Kruskal-Wallis rank sum test to determine the correlations between the continuous vari- ables and the outcome. If two of the selected continuous variables had high intercorrelation with a Pearson coefficient ≥0.9, one was removed.” This language may confuse the readers. The Kruskal-Wallis test is typically used to determine group differences, not correlations. For example, the mean income of those who died, and those who are alive. So you should write  “association” instead of correlation or you could write to study the differences in the continuous variables between groups.

2.       In Tables 2 and 3, explain at the foot of the table what the acronyms  PPV, NPV, and AUC mean.

3.       In Tables 2 and 3, you should compute 95% confidence intervals of  Sensitivity, Specificity PPV NPV. You can do that easily using the free software OpenEpi available at http://www.openepi.com/DiagnosticTest/DiagnosticTest.htm

Author Response

Reviewer: 3

I found the paper interesting and easy to read. Some small changes should be made.

Answer 0.

We would like to thank the Reviewer for all the positive remarks regarding our work. We assure the Reviewer that we have read carefully the suggestions from this decision letter and tried our best to improve the quality of the document accordingly.

Q1. The author say in line  155  “As the continuous parameters did not exhibit a normal distribution, we performed  Kruskal-Wallis rank sum test to determine the correlations between the continuous vari- ables and the outcome. If two of the selected continuous variables had high intercorrelation with a Pearson coefficient ≥0.9, one was removed.” This language may confuse the readers. The Kruskal-Wallis test is typically used to determine group differences, not correlations. For example, the mean income of those who died, and those who are alive. So you should write  “association” instead of correlation or you could write to study the differences in the continuous variables between groups.

Answer 1:

Thank you for a very good observation!

We modified the Methodology section accordingly:

“As the continuous parameters did not exhibit a normal distribution, we performed Kruskal-Wallis rank sum test to determine whether there are any associations between the continuous variables and the outcome.”

Q2. In Tables 2 and 3, explain at the foot of the table what the acronyms  PPV, NPV, and AUC mean.

Answer 2:

Thank you!

We added the explanations at the foot of the table, accordingly.

Q3. In Tables 2 and 3, you should compute 95% confidence intervals of  Sensitivity, Specificity PPV NPV. You can do that easily using the free software OpenEpi available at http://www.openepi.com/DiagnosticTest/DiagnosticTest.htm.

Answer 3:

Thank you for this suggestion. We added the 95% confidence intervals as the Reviewer recommended.